# Memory-Based Invariance Learning for Out-of-Domain Text Classification

**Chen Jia**[*]
SI-TECH Information Technology
Fudan University
jiachenwestlake@gmail.com

**Yue Zhang**
Westlake University
Westlake Institute for Advanced Study
zhangyue@westlake.edu.cn

## Abstract

We investigate the task of out-of-domain (OOD) text classification with the aim of extending a classification model, trained on multiple source domains, to an unseen target domain. Recent studies have shown that learning invariant representations can enhance the performance of OOD generalization. However, the inherent disparity in data distribution across different domains poses challenges for achieving effective invariance learning. This study addresses this issue by employing memory augmentations. Specifically, we augment the original feature space using key-value memory and employ a meta-learning-based approach to enhance the quality of the invariant representations. Experimental results on sentiment analysis and natural language inference tasks show the effectiveness of memory-based method for invariance learning, leading to state-of-the-art performance on six datasets.

## 1 Introduction

Text classification has made remarkable progress in recent years, thanks to the advancements in deep neural networks such as Transformer (Vaswani et al., 2017) and pretrained language models (PLMs) (Peters et al., 2018; Devlin et al., 2019; Brown et al., 2020). However, these learning systems heavily rely on the assumption that the training and test sets come from the same domain. When there is a significant discrepancy between the test domain (also known as the target domain) and the training domains (also known as source domains), the performance of traditional learning systems suffers significant declines (Blanchard et al., 2011; Muandet et al., 2013). Domain generalization (DG) aims to address this out-of-domain (OOD) problem, which is a practical and challenging issue, particularly when the labeled and unlabeled information of the target domains is unknown during the training phase.

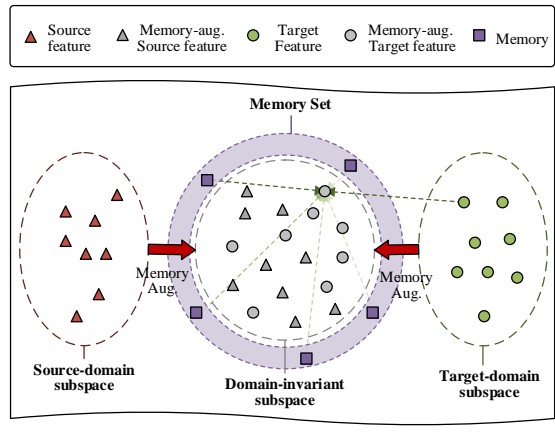

Figure 1: Memory-based invariant representation learning.

In this paper, we focus on a multi-source domain generalization setting where there are multiple source domains available for training. In recent years, domain-invariant representation learning has shown high effectiveness in multi-source DG (Ben-David et al., 2010; Ganin et al., 2016). Most existing approaches use the same model parameters across domains to construct a domain-shared feature space for domain-invariant representation learning (Li et al., 2018b; Albuquerque et al., 2019; Guo et al., 2020; Jia and Zhang, 2022b). However, the intrinsic distribution discrepancy across domains poses challenges for distribution matching in order to learn a domain-invariant feature space.

Inspired by recent work (Khandelwal et al., 2019, 2020; Zheng et al., 2021), which demonstrates that memory vectors can serve as rich feature augmentations for neural models, we propose to adopt memory augmentations to improve domain-invariant representation learning. As shown in Figure 1, the traditional parameter sharing mechanism produces distinct feature distributions between source domains (left) and target domains (right) due to the intrinsic domain discrepancy. To address this, we

---
[*]Corresponding author.

use memory augmentations to alleviate the discrepancy of feature distributions between source and target domains and improve the invariant feature distribution, constructing a domain-invariant feature subspace (middle).

To achieve this goal, we use a key-value memory network (Miller et al., 2016) to improve the Transformer model (Vaswani et al., 2017) by feature augmentation. Specifically, we employ a meta-learning strategy to learn memory augmentations for achieving the invariant representation distribution across domains. In each training episode, source domains are randomly split into the meta-target and meta-source domains to simulate domain shifts. Consequently, we propose a bi-level optimization objective to learn memory augmentations for domain invariance. The inner-loop objective is to minimize the meta-source risk w.r.t. the Transformer parameters, while the outer-loop objective is to minimize the domain discrepancy between the meta-source and meta-target samples w.r.t. the key-value memory, based on the optimized Transformer parameters from the inner-loop. As a result, after the meta-test phase, the memory augmentations improve the domain invariance between the source domain and unseen target domains.

We evaluate our method on sentiment analysis and natural language inference (NLI) tasks. The results show that the learned memory by bi-level optimization provides better augmentations to the feature representation compared with the traditional learning strategy. Our method achieves state-of-the-art results on six datasets, outperforming a range of strong baselines. To the best of our knowledge, we are the first to leverage a memory network for improving domain-invariant representation learning. The code will be released at https://github.com/jiachenwestlake/MIL.

## 2 Related Work

**Domain generalization.** In this work, we specifically focus on multi-source domain generalization (DG) (Blanchard et al., 2011; Muandet et al., 2013), which offers broader application opportunities compared to the single-source scenario (Qiao et al., 2020). With the advancements in deep neural networks (Li et al., 2017), DG has achieved promising results. Existing methods primarily aim to learn domain-invariant representations across source domains to enhance out-of-distribution (OOD) robustness, which have proven effective in objective

recognition tasks (Sun and Saenko, 2016; Li et al., 2018b; Arjovsky et al., 2019). However, these methods face challenges when there is a significant discrepancy between the source and target domains. The invariant classification model across source domains cannot easily adapt to unseen target domains. To address this issue, some studies in objective recognition (Li et al., 2018a; Balaji et al., 2018; Jia and Zhang, 2022a) and semantic parsing (Wang et al., 2021a) employ meta-learning-based approaches with episodic training strategies to improve model adaptation to domain shifts. In contrast to these works, we aim to learn explicit memory augmentations for domain transfer. In contrast, the meta-learner in our method aims to learn a domain-invariant feature space by learning a memory augmentation. The novelty of our work is reflected in the design of meta-learning objectives. The advantage of our design lies in the ability to leverage an additional memory network to learn more robust feature representations across domains.

Recently, there has been an increasing interest in DG for text classification. Ben-David et al. (2022) learn an example-based prompt for each instance for classification. In contrast, we focus on learning a more general memory augmentation that can address domain shifts comprehensively. Jia and Zhang (2022b) utilize a distribution alignment method to enhance domain invariance for DG. Tan et al. (2022a) adopt a memory-enhanced supervised contrastive learning method for DG. In comparison, we propose the use of key-value memory to explicitly augment feature representations and improve domain invariance in the feature space.

**Memory-based model adaptation.** The augmentation of neural networks with previous memory has proven effective for model adaptation in the testing phase (Santoro et al., 2016). Prior works improve network predictions using memory banks, which serve as a continuous cache for storing long-range contextual information (Khandelwal et al., 2019, 2020; Zhong et al., 2022). Memory banks have also shown utility for task adaptation (Santoro et al., 2016; Wang et al., 2021b). However, there are limited studies on memory-based cross-domain transfer. Existing works (Asghar et al., 2018; Zheng et al., 2021) rely on target-domain unlabeled data for domain transfer. However, these methods cannot be directly applied to DG since both labeled and unlabeled target information is

unknown during training. In contrast, we leverage memory to optimize the transferability from source domains to target domains through a meta-learning strategy.

To the best of our knowledge, only one existing memory-based work for DG refers to (Tan et al., 2022b), which leverages the memories of source-domain samples to augment contrasting features for computing supervised contrastive loss. Our work differs significantly from (Tan et al., 2022b). Firstly, our memory network is trainable, whereas they employ static source-domain banks that are not optimized during training. Secondly, we explicitly utilize memory as feature augmentation to enhance invariant representation learning, whereas they employ memory as contrasting features for computing the contrastive loss.

**Feature augmentation.** Previous studies have shown that model generalization can be improved by augmenting features through the mixing of feature vectors (Verma et al., 2019). In computer vision, prior works learn interpolation for semantic changes (Upchurch et al., 2017) or perturbs latent features with random noises using mix-up techniques (Zhou et al.; Li et al., 2021; Zhao et al., 2021). In contrast, we focus on learning memory augmentations to enhance domain invariance in the feature space.

## 3 Method

As illustrated in Figure 2, the proposed model comprises (a) a vanilla Transformer enhanced by (b) a key-value memory. Furthermore, (c) the output layer is responsible for text classification, while (d) the domain discriminators handle domain classification tasks.

The memory serves to enhance the feature representation and mitigate domain-specific feature distributions. To accomplish this, we employ a key-value memory bank that learns the appropriate feature augmentations (Section 3.1). To address domain shifts through memory augmentations, we introduce an episodic training strategy (Section 3.2). The training objective of the key-value memory can be formulated as bi-level optimization (Section 3.3). Lastly, we present the overarching meta-training and meta-test algorithms (Section 3.4).

### 3.1 Key-Value Memory-Augmented Network

We consider a key-value memory layer as a function $m : \mathbb{R}^d \to \mathbb{R}^d$, which can be trained end-

to-end by gradient backpropagation (Sukhbaatar et al., 2015). Following previous work (Miller et al., 2016; Lample et al., 2019), the overall structure of our memory layer consists of a query network and a value lookup table.

**Key-value memory.** Given a hidden state of one position from the previous layer $\mathbf{h} \in \mathbb{R}^d$, the query network acts as a function $q : \mathbf{h} \mapsto q(\mathbf{h}) \in \mathbb{R}^{d_q}$, mapping from a $d$-dimensional hidden vector into a latent query space with the dimensionality $d_q$. In this paper, $q(\cdot)$ is a linear mapping or a multi-layer perceptron to reduce the dimensionality of hidden space to a lower-dimensional query space for distance computation w.r.t. the keys.

Given a query $q(\mathbf{h})$ and a set of keys $\mathcal{K} = \{\mathbf{k}_1, \ldots, \mathbf{k}_{|\mathcal{K}|}\}$ that consists of $|\mathcal{K}|$ $d_q$-dimensional vectors, we first compute the dot-product similarity between the query and each key $\{\alpha_k\}_{k=1}^{|\mathcal{K}|}$. For each $k \in \{1, \ldots, |\mathcal{K}|\}$,

$$\alpha_k = \frac{\exp(q(\mathbf{h})^\top \mathbf{k}_k)}{\sum_{j=1}^{|\mathcal{K}|} \exp(q(\mathbf{h})^\top \mathbf{k}_j)} \quad (1)$$

Given a set of memory values $\mathcal{V} = \{\mathbf{v}_1, \ldots, \mathbf{v}_{|\mathcal{K}|}\}$ that consists of $|\mathcal{K}|$ $d_m$-dimensional vectors, the function of the key-value memory can be represented as a weighted sum of memory values:

$$m(\mathbf{h}) = \sum_{k=1}^{|\mathcal{K}|} \alpha_k \mathbf{v}_k \quad (2)$$

**Memory-augmented network.** We use the aggregated memory by key-value memory sublayer as feature augmentations for the original Transformer model to improve domain transfer. Particularly, we perform the feature augmentation through residual connection. Let $g : x \mapsto g(x) \in \mathbb{R}^d$ denote the Transformer model that mapping from an input text to a feature vector, we represent the memory-augmented network $g_m : x \mapsto g_m(x) \in \mathbb{R}^d$ as follows

$$g_m(x) = (1 - \lambda)g(x) + \lambda \cdot (m \circ g(x)), \quad (3)$$

where $\lambda$ represents the coefficient that balances the original features and augmented memory.

### 3.2 Episodic Training Procedure

Following Li et al. (2018a); Balaji et al. (2018), we leverage an episodic training procedure to simulate

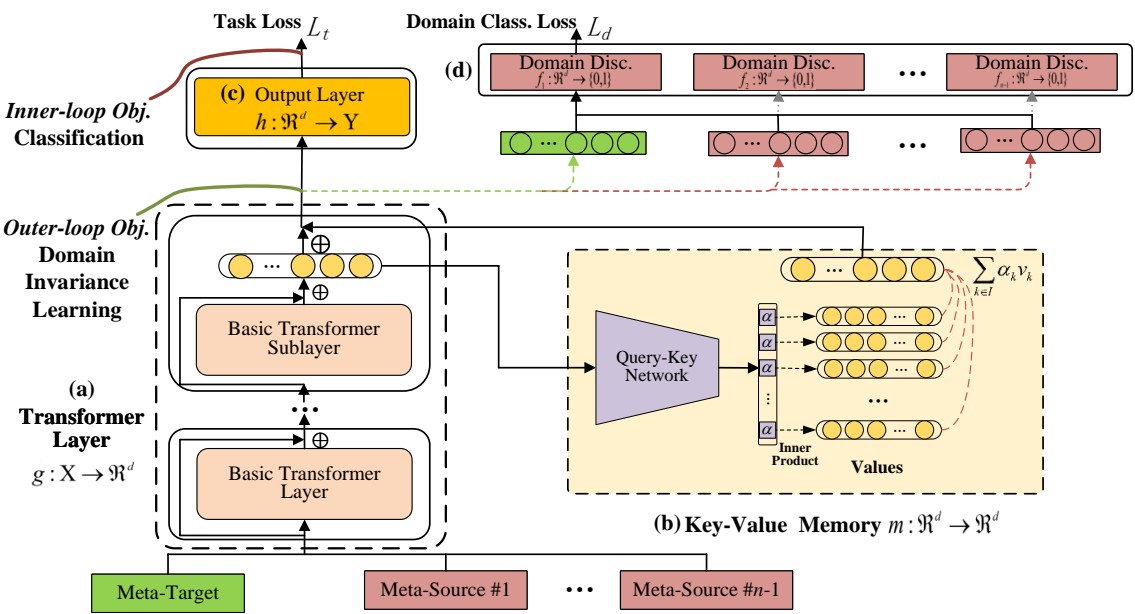

Figure 2: Overall structure of the proposed model.

**Algorithm 1** Episodic training process.

**Input:** Source domains $\mathcal{S} = \{S_1, S_2, \ldots, S_n\}$
**Input parameters:**.
**Output:** Optimized memory net $m^*$

1: **while** not converge or not reach stopping conditions **do**
2:     Randomly select a meta-target domain $\mathcal{D}_{te} \in \mathcal{S}$
3:     The meta-source domains are $\mathcal{D}_{tr} = \mathcal{S} - D_{te}$
4:     **for** $t \in \{1, \ldots, T\}$ **do**
5:         Sample mini-batch $D_{tr} \subset \mathcal{D}_{tr}$ and $D_{te} \subset \mathcal{D}_{te}$
6:         Optimize Transformer parameters $\boldsymbol{\theta}_g$ on $D_{tr}$
7:         Optimize the key-value memory parameters $\boldsymbol{\theta}_m$
         using the optimized Transformer parameters $\boldsymbol{\theta}_g^*$
8:     **end for**
9: **end while**

the domain shifts. Each episode can be viewed as a meta-task to learn how to learn a better key-value memory for tackling the domain shifts between source domains and the unseen target domains. In particular, the meta-task in this paper is specified as learning memory augmentations to improve the invariance of feature representations across domains.

A brief view of the episodic training process is shown in Algorithm 1. Given a set of source-domain training samples $\mathcal{S} = \{S_1, S_2, \ldots, S_n\}$, in each training episode, we first randomly select a meta-target domain and the rest serve as the meta-source domains (lines 2-3). Then, in each training iteration $t \in [T]$, we first optimize the Transformer model $g : \mathcal{X} \to \mathbb{R}^d$ parameterized by $\boldsymbol{\theta}_g$ and task output layer $h : \mathbb{R}^d \to \mathcal{Y}$ parameterized by $\boldsymbol{\theta}_h$ over the mini-batch of meta-source samples $D_{tr}$ (line 6). Then, we optimize the parameters of key-

value memory network $\boldsymbol{\theta}_m$ on the mini-batch of meta-target sample $D_{te}$ and meta-source samples $D_{tr}$ using the optimized Transformer parameters $\boldsymbol{\theta}_g^*$ (line 7).

### 3.3 Memory-Based Invariance Learning

Based on the episodic training process, we now describe in detail the optimization objectives w.r.t. the training samples and parameters.

**Domain-invariant representation learning objective.** Given $n$ training domains, we need $n$ domain discriminators to differ each domain from the other domains. To simplify the presentation, we use an unified function symbol $f_d$ to denote the domain discriminator between the meta-test (meta-target) data $\mathcal{D}_{te}$ and the meta-training (meta-source) data $\mathcal{D}_{tr}$. The domain classification objective can be represented as the binary cross-entropy loss:

$$\mathcal{L}_d = \sum_{x \in \mathcal{D}_{tr} \cup \mathcal{D}_{te}} \ell^{(\text{CE})}(f_d \circ g_m(x), \mathbb{I}_{[x \in \mathcal{D}_{te}]}) \quad (4)$$

The domain-invariant representaion learning solves the following minimax optimization objective w.r.t. the domain discriminator $f_d$ and key-value memory network $m$:

$$\max_{\boldsymbol{\theta}_m} \min_{\boldsymbol{\theta}_{f_d}} \mathcal{L}_d(\boldsymbol{\theta}_g, \boldsymbol{\theta}_m, \boldsymbol{\theta}_{f_d}; \mathcal{D}_{te}, \mathcal{D}_{tr}) \quad (5)$$

Theoretically, following Ben-David et al. (2010), the above minimax training objective aims to minimize the $\mathcal{H}$-divergence for obtaining the invariant representation between the meta-training (meta-source) and meta-test (meta-target) domains.

**Algorithm 2** Meta-training procedure.

---

**Input:** Source domains $\mathcal{S} = \{S_1, S_2, \ldots, S_n\}$
**Parameters (randomly initialized):** $\boldsymbol{\theta}_g, \boldsymbol{\theta}_h, \boldsymbol{\theta}_{f_d}, \boldsymbol{\theta}_m$
**Output:** Optimized memory net $m^*$

1: **while** not converge or not reach stopping conditions **do**
2:     Randomly select a meta-target domain $\mathcal{D}_{te} = S_d \in \mathcal{S}$
3:     The meta-source domains $\mathcal{D}_{tr} = \mathcal{S} - D_{te}$
4:     **for** $t \in \{1, \ldots, T\}$ **do**
5:         Sample mini-batch $D_{tr} \subset \mathcal{D}_{tr}$ and $D_{te} \subset \mathcal{D}_{te}$
6:         $L_t \leftarrow \mathcal{L}_t(\boldsymbol{\theta}_g, \boldsymbol{\theta}_m, \boldsymbol{\theta}_h; D_{tr})$       ▷ task obj.
7:         $\boldsymbol{\theta}_h \leftarrow \boldsymbol{\theta}_h - \eta \nabla_{\boldsymbol{\theta}_h} L_t$
8:         $\boldsymbol{\theta}'_g \leftarrow \boldsymbol{\theta}_g - \eta \nabla_{\boldsymbol{\theta}_g} L_t$
9:         $L_d \leftarrow \mathcal{L}_d(\boldsymbol{\theta}'_g, \boldsymbol{\theta}_m, \boldsymbol{\theta}_{f_d}; D_{tr}, D_{te})$   ▷ inv. obj.
10:       $\boldsymbol{\theta}_{f_d} \leftarrow \boldsymbol{\theta}_{f_d} - \eta \nabla_{\boldsymbol{\theta}_{f_d}} L_d$   ▷ min $L_d$ w.r.t. $f_d$
11:       $\boldsymbol{\theta}_m \leftarrow \boldsymbol{\theta}_m + \gamma \eta \nabla_{\boldsymbol{\theta}_m} L_d$   ▷ max $L_d$ w.r.t. $m$
12:     **end for**
13: **end while**
14: $\boldsymbol{\theta}_m^* \leftarrow \boldsymbol{\theta}_m$

---

**Task objective.** Let $\mathcal{D}_{tr}$ denote the meta-training data in each training episode, and $h : \mathbb{R}^d \to \mathcal{Y}$ denote the task classifier. We represent the task objective as the empirical risk on the meta-training data with the cross-entropy loss $\ell^{(\mathrm{CE})}(\cdot, \cdot)$:

$$\mathcal{L}_t = \sum_{(x,y) \in \mathcal{D}_{tr}} \ell^{(\mathrm{CE})}\big(h \circ g_m(x), y\big) \qquad (6)$$

**Bi-level optimization objective.** Given the meta-training data $\mathcal{D}_{tr}$ and meta-test data $\mathcal{D}_{te}$, we consider the following bi-level optimization objective for learning an optimized classification model $h^* \circ g_m^*$:

$$\boldsymbol{\theta}_m^* = \arg\max_{\boldsymbol{\theta}_m} \min_{\boldsymbol{\theta}_{f_d}} \mathcal{L}_d\big(\boldsymbol{\theta}_g^*, \boldsymbol{\theta}_m, \boldsymbol{\theta}_{f_d}; \mathcal{D}_{te}, \mathcal{D}_{tr}\big);$$

$$\boldsymbol{\theta}_g^*, \boldsymbol{\theta}_h^* = \underbrace{\arg\min_{\boldsymbol{\theta}_g, \boldsymbol{\theta}_h} \mathcal{L}_t(\boldsymbol{\theta}_g, \boldsymbol{\theta}_m, \boldsymbol{\theta}_h; \mathcal{D}_{tr})}_{\text{inner-loop objective}},$$

$$(7)$$

where the inner-loop optimization objective is the empirical task risk on the meta-training samples and the outer-loop optimization objective is the domain-invariant representation learning objective between the meta-target sample and meta-source samples.

### 3.4 Meta-Optimization Algorithm

We now design the full gradient-based algorithm to optimize the bi-level optimization objective in Eq. (7).

**Gradient update.** In the gradient-based optimization algorithm, the inner-loop optimization has $L$ gradient updating steps and the outer-loop optimization has $T$ gradient updating steps. Each

---

**Algorithm 3** Meta-test procedure.

---

**Input:** Source domains $\mathcal{S} = \{S_1, S_2, \ldots, S_N\}$
**Parameters (by meta-training):** $\boldsymbol{\theta}_m^*$
**Parameters (randomly initialized):** $\boldsymbol{\theta}_g, \boldsymbol{\theta}_h$
**Output:** Optimized model $h^* \circ g_m^*$

1: **while** not converge or not reach stopping conditions **do**
2:     Randomly select a training domain $\mathcal{D}_{tr} \in \mathcal{S}$
3:     **for** $t \in \{1, \ldots, T\}$ **do**
4:         Sample mini-batch $D_{tr} \subset \mathcal{D}_{tr}$
5:         $L_t \leftarrow \mathcal{L}_t(\boldsymbol{\theta}_g, \boldsymbol{\theta}_m^*, \boldsymbol{\theta}_h; D_{tr})$     ▷ task obj.
6:         $\boldsymbol{\theta}_h \leftarrow \boldsymbol{\theta}_h - \eta \nabla_{\boldsymbol{\theta}_h} L_t$   ▷ min $L_t$ w.r.t. $h$
7:         $\boldsymbol{\theta}_g \leftarrow \boldsymbol{\theta}_g - \eta \nabla_{\boldsymbol{\theta}_g} L_t$   ▷ min $L_t$ w.r.t. $g$
8:     **end for**
9: **end while**
10: $\boldsymbol{\theta}_g^* \leftarrow \boldsymbol{\theta}_g, \boldsymbol{\theta}_h^* \leftarrow \boldsymbol{\theta}_h$

---

gradient updating step in the inner-loop optimization is represented as:

Inner-loop opt. : for the $l^{\mathrm{th}} \in [L]$ step,

$$\boldsymbol{\theta}_g^{(l)} = \boldsymbol{\theta}_g^{(l-1)} - \eta \nabla_{\boldsymbol{\theta}_g} L_t(\boldsymbol{\theta}_g^{(l-1)}, \boldsymbol{\theta}_m, \boldsymbol{\theta}_h^{(l-1)}; D_{tr});$$

$$\boldsymbol{\theta}_h^{(l)} = \boldsymbol{\theta}_h^{(l-1)} - \eta \nabla_{\boldsymbol{\theta}_h} L_t(\boldsymbol{\theta}_g^{(l-1)}, \boldsymbol{\theta}_m, \boldsymbol{\theta}_h^{(l-1)}; D_{tr})$$

$$(8)$$

Each gradient updating step in the outer-loop optimization is represented as:

Outer-loop opt. : for the $t^{\mathrm{th}} \in [T]$ step,

$$\boldsymbol{\theta}_m^{(t)} = \boldsymbol{\theta}_m^{(t-1)} + \gamma \eta \nabla_{\boldsymbol{\theta}_m} L_d(\boldsymbol{\theta}_g^{(L)}, \boldsymbol{\theta}_m^{(t-1)}, \boldsymbol{\theta}_{f_d}^{(t-1)}; D_{te}, D_{tr});$$

$$\boldsymbol{\theta}_{f_d}^{(t)} = \boldsymbol{\theta}_{f_d}^{(t-1)} - \eta \nabla_{\boldsymbol{\theta}_{f_d}} L_d(\boldsymbol{\theta}_g^{(L)}, \boldsymbol{\theta}_m^{(t-1)}, \boldsymbol{\theta}_{f_d}^{(t-1)}; D_{te}, D_{tr}),$$

$$(9)$$

where $\eta$ represents the gradient updating rate and $\gamma$ represents the coefficient of gradient updating for the key-value memory network.

The full learning algorithm is a consequence of a meta-training procedure and a meta-test procedure, as shown in Algorithm 2 and Algorithm 3, respectively.

**Meta-training.** For each training episode, lines 4-12 in Algorithm 2 present $T$ iterations of parameter updating for the Transformer and key-value memory network. In particular, lines 6-8 present the inner-loop optimization by gradient updates on the parameters of Transformer and the task classifier. Then, lines 9-11 present the outer-loop optimization by gradient updates on the parameters of key-value memory network $\boldsymbol{\theta}_m$ based on the updated Transformer parameters $\boldsymbol{\theta}'_g$. As a result, the meta-training procedure preduces the optimized parameters of key-value memory network $\boldsymbol{\theta}_m^*$ (line 14).

**Meta-test.** Based on the learned parameters of key-value memory network $\boldsymbol{\theta}_m^*$ by the meta-training procedure, the meta-test procedure optimizes parameters of Transformer $\boldsymbol{\theta}_g$ and the task classifier $\boldsymbol{\theta}_h$ using all the source training data. In each iteration of lines 3-8 in Algorithm 3, the source training data are used to update the parameters of Transformer and task classifier by stochastic gradient descent (SGD). As a result, the meta-test procedure produces the optimized Transformer $\boldsymbol{\theta}_g^*$ and task classifier $\boldsymbol{\theta}_h^*$ (line 10).

After the meta-training and meta-test procedures, the optimized model $h^* \circ g_m^*$ can be used to make classification on the unseen target domain.

# 4 Experiments

We evaluate the proposed method on sentiment analysis and natural language inference (NLI) tasks.

## 4.1 Experimental Setup

**Datasets.** For the sentiment analysis task, we use Amazon Reviews (Blitzer et al., 2007) for leave-one-domain-out evaluation. This dataset comprises two classes (positive and negative) and four domains: book (B), DVD (D), electronics (E) and kitchen (K). Additionally, we include IMDB (Thongtan and Phienthrakul, 2019) and SST-2 (Socher et al., 2013) as test datasets for cross-dataset evaluation. For the NLI task, we employ a scaled-down version of MNLI (Ben-David et al., 2022)[1] for leave-one-domain-out evaluation. This dataset consists of three classes (entailment, neutral, contradiction) and five domains: fiction (F), government (G), slate (S), telephone (T) and travel (T'). Moreover, we use SNLI (Bowman et al., 2015) and SICK (Marelli et al., 2014) as test datasets for cross-dataset evaluation. Appendix A presents the statistics of the used datasets.

**Evaluation.** The evaluation methods include leave-one-domain-out evaluation (Gulrajani and Lopez-Paz, 2020) and cross-dataset evaluation (Jia and Zhang, 2022b). Specifically, we employ standard leave-one-domain-out evaluation on Amazon Reviews and MNLI, and cross-dataset evaluation on IMDB and SST-2 for sentiment analysis, as well as SNLI and SICK for NLI.

**Architecture and hyperparameters.** In all our experiments, we fine-tune RoBERTa$_{\text{BASE}}$ (Liu

---

[1] https://github.com/eyalbd2/PADA.

et al., 2019). We introduce a key-value memory sublayer after the 12$^{\text{th}}$ layer of RoBERTa$_{\text{BASE}}$. Further details regarding the model architecture and hyperparameters can be found in Appendix B.

## 4.2 Main Results

The results for sentiment analysis and NLI using RoBERTa$_{\text{BASE}}$ are presented in Table 1 and Table 2, respectively. Additionally, we include the results of another pre-trained language model (PLM), BERT$_{\text{BASE}}$, in Appendix C.1 to demonstrate the robustness of our approach.

Before investigating the performance of our method, we first analyze the challenges of OOD setting on the used text classification datasets by making comparisons to the in-domain setting. Compared with the in-domain results (*oracle*), directly testing on OOD data (*baseline*) shows a significant drop in performance. This indicates the difficulty of the used datasets for OOD evaluation.

The last four rows in Table 1 and Table 2 provide comparisons with four baselines. The notation "+ memory" indicates that the baseline model was augmented with key-value memory, similar to our approach, but without the bi-level optimization for invariance learning. "invariance learning (w/o memory)" refers to a method similar to the works by Li et al. (2018b); Albuquerque et al. (2019), which directly optimize domain invariance in the feature space without memory augmentations. The results indicate that "+ memory" does not significantly improve over the baseline, suggesting that simply integrating memory layers into the baseline model is insufficient for learning transferable information to address domain shifts. Although domain-invariant representation learning has been shown to be effective for out-of-distribution (OOD) objective recognition (Li et al., 2018b), "invariance learning (w/o memory)" only exhibits marginal improvements in our experiments. This suggests that traditional invariance learning methods face challenges in addressing OOD text classification. In comparison to these baselines, our method learns memory augmentations to improve domain invariance in the feature space and demonstrates significant enhancements in both sentiment analysis and NLI.

We compare our method with several state-of-the-art DG methods for text classification, most of which aim to achieve domain invariance across source domains. DEEP CORAL (Sun and Saenko,

| Method | Leave-one-domain-out on Amazon Reviews | | | | | Cross-dataset Evaluation | |
|---|---|---|---|---|---|---|---|
| Sources → Target | DEK→B | BEK→D | BDK→E | BDE→K | Avg. | Amazon→IMDB | Amazon→SST-2 |
| supervised learning (*oracle*) | 95.0 | 94.3 | 95.3 | 96.4 | 95.3 | 94.9 | 93.4 |
| DEEP CORAL (Sun and Saenko, 2016) | 91.9 | 91.3 | 90.9 | 93.5 | 91.9 | 89.8 | 87.6 |
| IRM (Arjovsky et al., 2019) | 92.3 | 91.2 | 91.9 | 94.5 | 92.5 | 89.0 | 86.7 |
| PADA (Ben-David et al., 2022) | 86.8 | 86.9 | 89.0 | 92.6 | 88.8 | - | - |
| PDA (Jia and Zhang, 2022b) | 92.9 | 92.2 | 93.3 | 94.8 | 93.3 | 92.1 | 91.3 |
| M-SCL (Tan et al., 2022a) | 92.3 | 91.2 | 93.7 | 93.4 | 92.7 | - | - |
| RoBERTa_BASE (*baseline*) | 91.5 | 90.5 | 92.2 | 93.7 | 92.0 | 90.1 | 88.3 |
| + memory | 92.0 | 91.5 | 91.8 | 93.2 | 92.1 | 90.5 | 88.7 |
| invariance learning (w/o memory) | 92.2 | 90.7 | 92.5 | 94.2 | 92.4 | 91.4 | 89.2 |
| **our method** | **93.5** | **92.8** | **94.7** | **95.2** | **94.0**$^\dagger$ | **93.5**$^\dagger$ | **92.4**$^\dagger$ |

Table 1: Macro-$F_1$ on sentiment analysis. The best and second best scores of each column are marked in bold and underline, respectively. $^\dagger$ indicates statistical significance with $p < 0.05$ by $t$-test compared to all baselines.

| Method | Leave-one-domain-out on MNLI | | | | | | Cross-dataset Evaluation | |
|---|---|---|---|---|---|---|---|---|
| Sources → Target | GSTT'→F | FSTT'→G | GFTT'→S | GSFT'→T | GSTF'→T' | Avg. | MNLI→SNLI | MNLI→SICK |
| supervised learning (*oracle*) | 83.2 | 88.3 | 81.8 | 82.7 | 86.4 | 84.5 | 88.5 | 90.3 |
| DEEP CORAL (Sun and Saenko, 2016) | 77.6 | 76.3 | 78.2 | 75.3 | 78.2 | 77.1 | 77.3 | 57.0 |
| IRM (Arjovsky et al., 2019) | 78.1 | 75.2 | 79.4 | 76.2 | 79.2 | 77.6 | 76.2 | 58.7 |
| PADA (Ben-David et al., 2022) | 76.4 | 83.4 | 76.9 | 78.9 | 82.5 | 79.6 | - | - |
| PDA (Jia and Zhang, 2022b) | 80.8 | 85.8 | 79.7 | 79.4 | 83.0 | 81.7 | 79.3 | 62.0 |
| RoBERTa_BASE (*baseline*) | 79.5 | 80.2 | 79.7 | 76.8 | 80.0 | 79.2 | 78.1 | 61.5 |
| + memory | 79.6 | 80.7 | 79.2 | 77.0 | 81.2 | 79.5 | 78.6 | 60.8 |
| invariance learning (w/o memory) | 80.2 | 83.2 | 77.4 | 78.2 | 81.3 | 80.1 | 77.0 | 61.3 |
| **our method** | **81.2** | **86.3** | **80.5** | **80.4** | **84.6** | **83.0**$^\dagger$ | **82.3**$^\dagger$ | **65.7**$^\dagger$ |

Table 2: Macro-$F_1$ on NLI. The best and second best scores of each column are marked in bold and underline, respectively. $^\dagger$ indicates statistical significance with $p < 0.05$ by $t$-test when compared to all baselines.

2016) learns domain-invariant feature representations by optimizing second-order statistics over feature states. IRM (Arjovsky et al., 2019) further considers the intrinsic relationship between feature representation and labeling prediction to tackle domain shifts. PDA (Jia and Zhang, 2022b) simultaneously learns domain invariance for both feature representation and predicted probability. M-SCL (Tan et al., 2022a) employs a supervised contrast learning method with memory augmentations to increase the contrasting examples.

To ensure fair comparison, we reproduce M-SCL on sentiment analysis using RoBERTa_BASE, while the results of the other methods are taken from the literature that uses RoBERTa_BASE. For leave-one-domain-out evaluation, our method outperforms all the compared methods by 0.7% $F_1$ and 1.3% $F_1$ on the Amazon Reviews and MNLI datasets, respectively. In terms of cross-dataset evaluation, our method achieves over 1.0% $F_1$ improvement on two sentiment analysis datasets and approximately 3.0% $F_1$ improvement on two NLI datasets compared to the other methods. These results demonstrate the superiority of employing meta-learning

| Method | #Params | Amazon | MNLI | Avg. |
|---|---|---|---|---|
| RoBARTa_BASE | 108M | 92.0 | 79.2 | 85.6 |
| + memory | 113M | 92.1 | 79.5 | 85.8 |
| + FFN | 113M | 91.0 | 77.5 | 84.3 |
| + self-attn + FFN | 115M | 91.5 | 78.6 | 85.1 |
| **our method** | 113M | **94.0** | **83.0** | **88.5**$^\dagger$ |

Table 3: Macro-$F_1$ on two datasets. $^\dagger$ indicates statistical significance with $p < 0.05$ by $t$-test compared to all baselines.

to acquire transferable memory for domain generalization.

### 4.3 Analysis

**Effects of Additional Parameters.** Our method utilizes an additional key-value memory layer and includes approximately 4.8M more parameters compared to the RoBARTa_BASE baseline model. To ensure a fair comparison in terms of parameter size, we consider three additional baselines: (i) "+ memory uses the same key-value memory as our method but does not employ our invariance learning technique; (ii) "+ FFN" adds a feed-forward network (FFN) to the RoBARTa_BASE

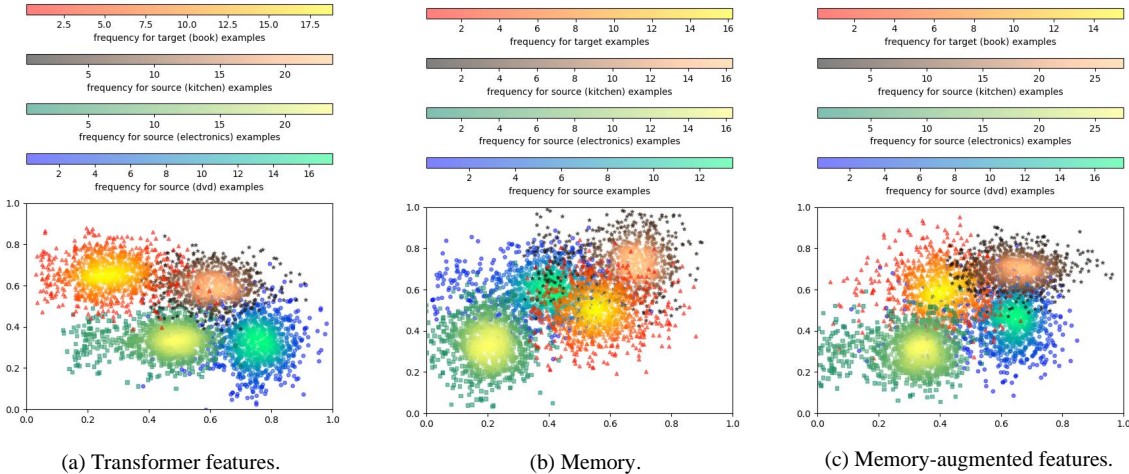

(a) Transformer features.     (b) Memory.     (c) Memory-augmented features.

Figure 3: Feature representation on the Amazon Reviews dataset. We visualize the representation of the target *book* domain and the source domains using Gaussian kernel density estimation and PCA dimensional reduction into $[0, 1] \times [0, 1]$. We compare (a) Transformer features, (b) memory and (c) memory-augmented features.

model; and (iii) "+ self-attn + FFN" incorporates both a self-attention layer and an FFN on top of the RoBARTa$_{\mathrm{BASE}}$ model. Although these three baselines have a similar number of parameters as our method, they do not yield significant improvements in performance. This observation indicates that merely increasing the parameter size with additional layers does not enhance out-of-distribution (OOD) text classification, thus demonstrating the effectiveness of our memory-based invariance learning method.

**Visualization.** We adopt t-SNE (Van der Maaten and Hinton, 2008) to visualize the feature representations, as shown in Figure 3. From Figure 3 (a), we can observe that the Transformer features of the target domain exhibit a distinctly different distribution compared to those of the source domains. However, with the aid of memory augmentations, Figure 3 (c) shows a smaller distance between the features of the target domain and those of the source domains. Interestingly, the memory distribution in Figure 3 (b) reveals a strong domain specificity across different domains. These findings demonstrate that our method is capable of effectively learning memory augmentations for different domains, thereby achieving domain invariance in the feature space.

**Invariant representation learning.** We adopt the $\mathcal{A}$-distance (Ben-David et al., 2006) to measure the distance of feature distributions between the target domain and the source domains using three sentiment analysis datasets. As depicted in Figure 4, incorporating the key-value memory over

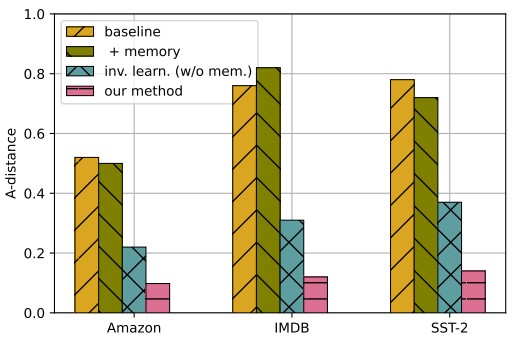

Figure 4: $\mathcal{A}$-distance between the target domain and sources domains on three sentiment analysis datasets. The results for NLI are illustrated in Appendix C.2.

the RoBARTa$_{\mathrm{BASE}}$ model without employing the invariance learning strategy barely improves the $\mathcal{A}$-distance. In contrast, the traditional invariant representation learning approach proves effectiveness in reducing the target-source domain $\mathcal{A}$-distance. Furthermore, our method further optimizes the $\mathcal{A}$-distance to a much greater extent, which suggests that the memory learned by our method contributes to the invariance of feature representations.

**Effects of memory learning.** As demonstrated in Figure 5, the development results for both the Amazon Reviews and MNLI datasets show a significant increase as the memory size increases from 128 to 1,024. This observation indicates that a larger memory bank size encompasses richer features, allowing for accurate memory augmentations in generating domain-invariant representations. However, the

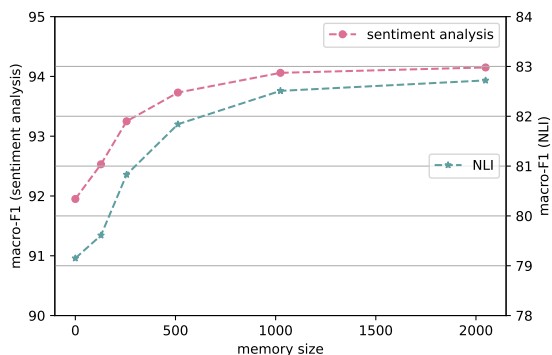

Figure 5: Effects of memory size. Average results on the dev sets of Amazon Reviews and MNLI.

magnitude of performance improvement tends to diminish as the memory size continues to increase, especially when the memory size exceeds 1,024. In our experiments, we choose a memory size of 1,024 to strike a balance between performance and model size. Additionally, we also analyze the effects of memory optimization in Appendix C.3.

## 5 Conclusion

We have conducted an investigation into a memory-based approach for domain generalization (DG). Our study involves the integration of a key-value memory network into the Transformer model, and the proposal of a meta-learning algorithm that incorporates an episodic training strategy to effectively learn transferable memory for addressing domain shifts. The results obtained from experiments conducted on sentiment analysis and natural language inference tasks demonstrate the significant enhancement in transferability of the source-domain model through the usage of the memory unit. Additionally, our approach achieves state-of-the-art performance on six different datasets.

## Limitations

Our method only applies the BASE-level pretrained language models, such as RoBERTa$_{\text{BASE}}$ and BERT$_{\text{BASE}}$. The recently developed large-scale pretrained language models, such as RoBERTa$_{\text{LARGE}}$ and GPT (Brown et al., 2020) have shown strong performances on classification and generatioin tasks. Due to resource limitations, we leave such large-model results in future work.

## Ethics Statement

We agree with the statements of the ACL Code of Ethics and declare that this submission follows the submission policies of ACL.

## Acknowledgments

We thank the anonymous reviewers for their helpful comments and suggestions. We gratefully acknowledge funding from the National Natural Science Foundation of China (NSFC No. 61976180) and the Zhejiang Province Key Project 2022SDX-HDX0003.

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

# A   Statistical Details of the Used Datasets

| Dataset | Domain | Train (src) | Dev (src) | Test (tgt) |
|---|---|---|---|---|
| **Sentiment Analysis** | | | | |
| Amazon | *book* (B) | 1.6K | 0.4K | 0.4K |
| | *DVD* (D) | 1.6K | 0.4K | 0.4K |
| | *electronics* (E) | 1.6K | 0.4K | 0.4K |
| | *kitchen* (K) | 1.6K | 0.4K | 0.4K |
| IMDB | *movie* | - | - | 25K |
| SST-2 | *movie* | - | - | 1.8K |
| **NLI** | | | | |
| MNLI | *fiction* (F) | 2.5K | 2.0K | 2.0K |
| | *government* (G) | 2.5K | 1.9K | 1.9K |
| | *slate* (S) | 2.6K | 2.0K | 2.0K |
| | *telephone* (T) | 2.8K | 2.0K | 2.0K |
| | *travel* (T') | 2.5K | 2.0K | 2.0K |
| SNLI | general | - | - | 9.8K |
| SICK | *image&video* | - | - | 0.5K |

Table 4: Statistics of the used datasets.

For the sentiment analysis task, we use Amazon Reviews (Blitzer et al., 2007), which comprises two classes (positive and negative) and four domains: book (B), DVD (D), electronics (E) and kitchen (K). Additionally, we include IMDB (Thongtan and Phienthrakul, 2019) and SST-2 (Socher et al., 2013) as test datasets for cross-dataset evaluation. For the NLI task, we employ a scaled-down version of MNLI (Ben-David et al., 2022), which consists of three classes (entailment, neutral, contradiction) and five domains: fiction (F), government (G), slate (S), telephone (T) and travel (T'). Moreover, we use SNLI (Bowman et al., 2015) and SICK (Marelli et al., 2014) as test datasets for cross-dataset evaluation. Table 4 presents the statistics of the used datasets.

# B   Details on Architecture and Hyperparameters

We utilize RoBERTa$_{\mathrm{BASE}}$ (Liu et al., 2019) as the primary Pretrained Language Models (PLMs) in our study, following the OpenPrompt framework (Ding et al., 2022), for two text classification tasks. The entire model was trained for up to 20 epochs, with a mini-batch size of 32 sentences applied across all datasets. Optimization was performed using AdamW with an initial learning rate set to $1e^{-5}$, a weight decay rate of 0.01, and warm-up steps of 500. We incorporated a key-value memory layer after the 12-th layer of RoBERTa$_{\mathrm{BASE}}$. This memory layer was added exclusively to the position used for classification, such as [MASK]

during prompting or [CLS] during traditional fine-tuning. To ensure balanced features, we selected a coefficient $\gamma$ of 0.5 for our experiments. For each key-value memory network, the hidden size of keys was set to 256, and the default number of values was 1024. Following the methodology of Lample et al. (2019), we employed a multi-head query-key attention mechanism with four heads. The total parameter count of the memory layers was approximately 4.8M, significantly smaller compared to the 108M total parameters of RoBERTa$_{\mathrm{BASE}}$.

# C   Additional Results

## C.1   Results based on BERT$_{\mathrm{BASE}}$

The results obtained using BERT$_{\mathrm{BASE}}$ are consistent with the main findings presented in Table 1 and Table 2, as illustrated in Table 5 and Table 6. The baseline models, namely "+ memory" and "invariance learning (w/o memory)", either show minimal or no significant improvement compared to the baseline model. In contrast, our method demonstrates superior performance in both sentiment analysis and NLI tasks, surpassing these baseline models. This indicates the robustness of our approach across different pre-trained language models (PLMs).

## C.2   Invariant Representation Learning for NLI

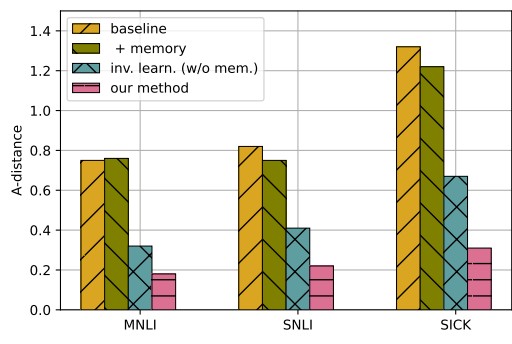

Figure 6: $\mathcal{A}$-distance between the target domain and sources domains on three NLI datasets.

We adopt the $\mathcal{A}$-distance (Ben-David et al., 2006) to measure the distance of feature distributions between the target domain and the source domains on three NLI datasets. As depicted in Figure 6, incorporating the key-value memory over the RoBARTa$_{\mathrm{BASE}}$ model without employing the invariance learning strategy barely improves the

| Method | Leave-one-domain-out on Amazon Reviews | | | | | Cross-dataset Evaluation | |
| --- | --- | --- | --- | --- | --- | --- | --- |
| Sources → Target | DEK→B | BEK→D | BDK→E | BDE→K | Avg. | Amazon→IMDB | Amazon→SST-2 |
| supervised learning (*oracle*) | 92.6 | 92.4 | 91.1 | 93.7 | 92.5 | 92.2 | 90.7 |
| DEEP CORAL (Sun and Saenko, 2016) | 88.2 | 87.8 | 88.2 | 89.7 | 88.5 | 85.8 | 85.7 |
| IRM (Arjovsky et al., 2019) | 85.8 | 89.4 | 87.6 | 90.5 | 88.3 | 84.8 | 84.2 |
| PDA (Jia and Zhang, 2022b) | 90.2 | 89.7 | 90.8 | 91.6 | 90.6 | 88.5 | 86.8 |
| BERT$_{BASE}$ (*baseline*) | 89.1 | 88.7 | 88.2 | 90.8 | 89.2 | 86.0 | 85.1 |
| + memory | 90.0 | 88.2 | 88.7 | 89.6 | 89.1 | 85.8 | 86.0 |
| invariance learning (w/o memory) | 89.8 | 88.7 | 91.0 | 90.6 | 90.0 | 88.6 | 86.2 |
| **our method** | **90.6** | **90.7** | **91.5** | **92.8** | **91.4**[†] | **89.2**[†] | **87.8**[†] |

Table 5: Macro-F$_1$ on sentiment analysis based on BERT$_{BASE}$. The best and second best scores of each column are marked in bold and underline, respectively. [†] indicates statistical significance with $p < 0.05$ by $t$-test compared to all baselines.

| Method | Leave-one-domain-out on MNLI | | | | | | Cross-dataset Evaluation | |
| --- | --- | --- | --- | --- | --- | --- | --- | --- |
| Sources → Target | GSTT'→F | FSTT'→G | GFTT'→S | GSFT'→T | GSTF→T' | Avg. | MNLI→SNLI | MNLI→SICK |
| supervised learning (*oracle*) | 80.4 | 82.6 | 76.3 | 78.5 | 81.7 | 80.6 | 83.0 | 89.6 |
| DEEP CORAL (Sun and Saenko, 2016) | 75.7 | 74.6 | 73.1 | 74.0 | 76.3 | 74.7 | 65.5 | 56.3 |
| IRM (Arjovsky et al., 2019) | 74.2 | 75.8 | 71.8 | 73.7 | 75.1 | 74.1 | 65.8 | 57.0 |
| PDA (Jia and Zhang, 2022b) | 75.2 | 76.8 | 72.8 | 74.6 | 77.8 | 75.4 | 67.6 | 60.4 |
| BERT$_{BASE}$ (*baseline*) | 74.8 | 72.8 | 72.5 | 72.9 | 74.7 | 73.5 | 64.8 | 55.2 |
| + memory | 73.6 | 73.8 | 72.2 | 72.0 | 75.8 | 73.5 | 65.7 | 53.2 |
| invariance learning (w/o memory) | 76.8 | 75.4 | 72.7 | 74.8 | 78.2 | 75.6 | 66.7 | 61.0 |
| **our method** | **77.0** | **78.2** | **73.6** | **76.2** | **78.6** | **76.7**[†] | **70.5**[†] | **61.2**[†] |

Table 6: Macro-F$_1$ on NLI based on BERT$_{BASE}$. The best and second best scores of each column are marked in bold and underline, respectively. [†] indicates statistical significance with $p < 0.05$ by $t$-test when compared to all baselines.

$\mathcal{A}$-distance. In contrast, the traditional invariant representation learning approach proves effective in reducing the target-source domain $\mathcal{A}$-distance. Furthermore, our method further optimizes the $\mathcal{A}$-distance to a much greater extent, which suggests that the memory learned by our method contributes to the invariance of feature representations.

## C.3 Effects of Memory Optimization

the learning rate for memory values ranges from 0 to $1e^{-3}$. When the learning rate is set to 0, the key-value memory network remains untrained, thus failing to produce appropriate memory augmentations. As a consequence, the results are noticeably lower than those of the baseline model without memory augmentations. As the learning rate gradually increases, the results improve with minor fluctuations, ultimately reaching a plateau when the learning rate reaches a sufficiently high value. This indicates that optimizing the key-value memory network facilitates the performance of OOD text classification.

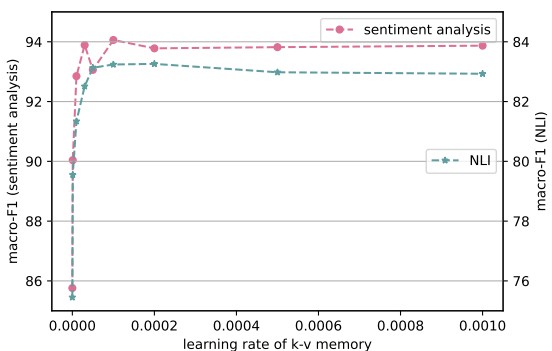

Figure 7: Effects of memory optimization. Average results on the dev sets of Amazon Reviews and MNLI.

Figure 7 presents the results obtained from the Amazon Reviews dev set and the MNLI dev set, as