# OpenReview forum: "Memory-Based Invariance Learning for Out-of-Domain Text Classification"
_EMNLP/2023/Conference — EMNLP 2023 Main_

### Official Review · Reviewer_545j · 2023-08-04

**Soundness:** 3

**Excitement:**

4: Strong: This paper deepens the understanding of some phenomenon or lowers the barriers to an existing research direction.

**Paper Topic And Main Contributions:**

This paper titled "Memory-Based Invariance Learning for Out-of-Domain Text Classification" is about a novel approach to improve out-of-domain text classification.

The paper proposes a memory-based invariant representation learning method that utilizes a memory network. This memory network helps in constructing domain-invariant representations.

The method is evaluated on sentiment analysis and natural language inference tasks and achieves state-of-the-art results on six datasets.
The paper also compares the method to several baselines and analyzes the effects of memory size and memory optimization.

Overall, the paper presents a novel approach to domain-invariant representation learning that significantly improves the accuracy of out-of-domain text classification tasks.

**Reasons To Accept:**

1) Memory-Based Invariant Representation Learning: The paper introduces a novel memory-based approach for domain-invariant representation learning. The proposed method constructs domain-invariant representations across multiple source domains.

2) Improved Out-of-Domain Text Classification: The memory-based approach significantly enhances the accuracy of out-of-domain text classification tasks, specifically sentiment analysis and natural language inference. The method achieves state-of-the-art results on six datasets, surpassing several strong baselines.

3) Comparison and Analysis: The paper compares the proposed memory-based method with traditional learning strategies and analyzes the effects of memory size and memory optimization. The results show that the memory-based approach provides better augmentations to the feature representations and contributes to the invariance of feature representations.

**Reasons To Reject:**

The underlying base encoder used in the experiments of the paper is not specified. It is necessary to conduct experiments to discuss whether different base models are sensitive to different out-of-domain methods.

**Reproducibility:**

3: Could reproduce the results with some difficulty. The settings of parameters are underspecified or subjectively determined; the training/evaluation data are not widely available.

**Reviewer Confidence:**

3: Pretty sure, but there's a chance I missed something. Although I have a good feel for this area in general, I did not carefully check the paper's details, e.g., the math, experimental design, or novelty.

---

> ### Author Rebuttal · Authors · 2023-08-28
>
> ### Dear Reviewer 545j,
>
> We thank your efforts in giving detailed and constructive comments and appreciate your support of our work. We hope our response can successfully address your concerns.
>
> > **Weakness 1**: The underlying base encoder used in the experiments of the paper is not specified. It is necessary to conduct experiments to discuss whether different base models are sensitive to different out-of-domain methods.
>
> **Response:** Thanks for your suggestion. To assess the sensitivity of OOD methods to different basic encoders, we further evaluate their performance using alternative encoders such as **BiLSTM** and **CNN** in addition to RoBERTa and BERT (refer to Table 1&2 and Table 5&6, respectively). The hyperparameters for text classification using BiLSTM and CNN are adopted from [1] and [2], respectively. **Notably, our findings exhibit similar trends to the results obtained with RoBERTa and BERT, providing further evidence of the effectiveness of our approach. Additionally, our method attains superior results on all six datasets, demonstrating its efficacy in OOD text classification across various base encoding models, including LSTM and CNN-based architectures.**
>
> | Method | Amazon | IMDB | SST-2 | MNLI | SNLI | SICK |
> | ---------- | ---------- | ---------- | ---------- | ---------- | ---------- |---------- |
> |BiLSTM (*baseline*) | 80.1 | 75.6 | 74.8 | 65.1 | 64.8 | 52.6 |
> | + memory | 79.6 | 75.2 | 75.3 | 65.5 | 63.7 | 53.9 |
> | inv. learn. (w/o memory)  | 81.4 | 76.7| 77.2 | 66.2| 65.5 | 51.8 |
> | **our method**  | **82.2** | **77.4** | **77.5** | **67.1** | **66.7** | **54.3** |
>
> | Method | Amazon | IMDB | SST-2 | MNLI | SNLI | SICK |
> | ---------- | ---------- | ---------- | ---------- | ---------- |---------- | ---------- |
> |CNN (*baseline*) | 78.2 | 74.0 | 70.3 | 64.2 | 58.7 | 54.2 |
> | + memory | 76.7 | 74.8 | 70.8 | 65.2 | 60.2 | 52.8 |
> | inv. learn. (w/o memory)  | 79.2 | 75.3 | 72.3 | 64.7 | 61.7 | 54.7  |
> | **our method**  | **80.1** | **75.8** | **73.5** | **65.8** | **62.5** |  **55.4**  |
>
> [1] Adversarial Multi-task Learning for Text Classification. Liu et al., 2017
>
> [2] Understanding Convolutional Neural Networks for Text Classification. Jacovi et al., 2018.

---

### Official Review · Reviewer_VVu8 · 2023-08-05

**Typos Grammar Style And Presentation Improvements:** Please refer to "Reasons To Reject" f…
**Soundness:** 3

**Excitement:**

3: Ambivalent: It has merits (e.g., it reports state-of-the-art results, the idea is nice), but there are key weaknesses (e.g., it describes incremental work), and it can significantly benefit from another round of revision. However, I won't object to accepting it if my co-reviewers champion it.

**Paper Topic And Main Contributions:**

This paper studies the problem of out-of-domain (OOD) text classification and focuses on domain generalization (DG). One main challenge for DG is the distribution disparity across different domains, which can make it difficult to learn the effective invariant representation. Hence the main contribution of this paper is to augment the transformer with a key-value memory network to meta-learn the domain-invariant representation. The authors evaluate their method on sentiment analysis and natural language inference tasks which exhibit considerable domain variations, and the experiments show the effectiveness of their method.

**Questions For The Authors:**

Question A: What are the keys in the key-value memory network (L207)?

Question B: What is the memory size in Figure 5? Is it the number of keys?

Question C: How can we reach the claim in L476-479 from Figure 3?

**Reasons To Accept:**

The results of the proposed method are very good, and the ablation experiments show the effectiveness of memory network and invariant representation learning.

**Reasons To Reject:**

1) This paper fails to explain the motivation or provide the sufficient rationale for their method designs in many places. For example, why the key-value memory is used and why meta-learning is adopted to learn the invariant representation instead of other approaches? Why minimax the domain classification objective can learn the invariant representation (i.e., how domain discriminators help)? In the experimental section, why the tasks of sentiment analysis and natural language inference are chosen to study domain generalization; is it to follow prior works or because these tasks have diverse domains? The lack of motivation may raise doubts about the necessity, effectiveness (and novelty) of these design choices. Explaining why doing this is better than simply listing what is done.

2) This paper is not very well-written and organized. For example, Section 3 lacks clarity, and it contains typos (e.g., "f_d" instead of "f_g" in equation (5)) and confusing notation that  seems to influence the understanding of the algorithms. Moreover, overloading a section with excessive content can lead to confusion and make it hard for readers to follow the logic flow of ideas. It would be better if each section can address a distinct aspect of the work, e.g., a separate section of task formulation from the current Section 3. Last but not least, it is confusing to mention "meta-test" in Section 3.4: is it a different concept from the inner-loop in Section 3.3?

3) The illustration of paper can be more informative. For example, figure 2 can explain the component (d) is only used during training to help the memory network to learn the invariant representation; the current version may make the readers misunderstand it will also work in inference.

4) Both meta-learning and memory networks have been used before in addressing the out-of-domain text classification. What is the most motivating thing (i.e., novelty) about this work? It would be better if the authors can explain why the approach is a significant contribution over previous methods, and what the implication is for future work.

**Reproducibility:**

4: Could mostly reproduce the results, but there may be some variation because of sample variance or minor variations in their interpretation of the protocol or method.

**Reviewer Confidence:**

3: Pretty sure, but there's a chance I missed something. Although I have a good feel for this area in general, I did not carefully check the paper's details, e.g., the math, experimental design, or novelty.

---

> ### Author Rebuttal · Authors · 2023-08-26
>
> ### Dear Reviewer VVu8,
>
> Thank you for taking the time to read our paper and giving constructive comments. We hope your concerns can be addressed by our point-to-point responses.
>
> > **Weakness 1:** This paper fails to explain the motivation or provide the sufficient rationale for their method designs in many places.
>
> We make responses to each concept of this concern as follows.
>
> > **Weakness 1-1:** Why the key-value memory is used and why meta-learning is adopted to learn the invariant representation instead of other approaches?
>
> **Response:** Thanks for your valuable comments. Here, we explain the motivation in **two folds**.
>
> Firstly, we use the **K-V memory** as **feature augmentation** to improve the **invariant feature representation** across domains. Compared with the traditional methods that learn a shared feature space across domains [1], we use memory augmentations to **alleviate the discrepancy of feature distributions** between source and target domains and **improve the invariant feature distribution**, as shown in Fig 1.
>
> Secondly, to enhance memory-based invariance learning, we employ a **meta-learning strategy**. **Meta-learning is chosen over alternative invariance learning methods such as [2, 3] due to the inherent challenge of domain generalization, where the target domain remains unseen during training**. Consequently, it becomes impractical to explicitly optimize invariant features between the source and target domains, as traditionally done in invariance optimization methods [2, 3]. Therefore, we adopt a meta-learning approach within an episodic training process. We now elaborate on the meta-learning procedure.
>
> In each episode of meta-training, we randomly partition the source domains into meta-source and meta-target samples, simulating the domain shifts encountered in the testing phase. Our proposed bi-level optimization method addresses the discrepancy between the meta-source and meta-target samples (outer-loop) while optimizing the classification task using the meta-source samples (inner-loop). Subsequent to the meta-training phase, we proceed to a meta-test phase, where we optimize a classification task using the source samples. Through the utilization of the meta-learning strategy, we attain domain invariance between the test domain and the source domains.
>
> [1] Deeper, broader and artier domain generalization. Li et al., 2017.
>
> [2]  A theory of learning from different domains. Ben-David et al., 2010.
>
> [3] Generalizing to unseen domains via distribution matching. Albuquerque et al., 2019.
>
> > **Weakness 1-2**: Why minimax the domain classification objective can learn the invariant representation (i.e., how domain discriminators help)?
>
> **Response:**  We follow the previous work [2] (Thm. 1) to minimize the **$\mathcal{H}$-divergence** for obtaining the invariant representation. The $\mathcal{H}$-divergence between the meta-source domain $D_{tr}$ and meta-target domain ${D}_{te}$ can be represented as:
>
> $d_{\mathcal{H}}(D_{tr}, D_{te}) =  \sup_{f_d} \vert  Pr_{D_{tr}}(I(f_d)) -   Pr_{D_{te}}(I(f_d))  \vert$,
> where $I(f_d)$ represents a domain indicator such that $x \in I(f_d) \iff f_d(x) = 1$.
>
> Then, using Lemma 2 in [2], we obtain that
>
> $d_{\mathcal{H}} (D_{tr}, D_{te}) = 1 - 2\min_{f_d} L_d (f_d) $, where $L_d (f_d)$ (Eq. (4)) represents the empirical error for a domain discriminator $f_d$ to distinguish samples from $D_{tr}$ and $D_{te}$, i.e., $f_d (x) = 0$ if $x \in D_{tr}$ and $f_d (x) = 1$ if $x \in D_{te}$.
>
> Thus, the minimax objective $\max_{m}\min_{f_d} L_d$ $\iff$ $\min_{m} d_{\mathcal{H}}(g_m(D_{tr}), g_m(D_{te}))$. This indicates that **minimax the domain classification objective is equivalent to minimizing the $\mathcal{H}$-divergence** between $D_{tr}$ and $D_{te}$ w.r.t. the memory augmentation $m$, which can be viewed as learning the invariant representation across domains in the previous work [2,3].
>
> > **Weakness 1-3**: In the experimental section, why the tasks of sentiment analysis and natural language inference are chosen to study domain generalization; is it to follow prior works or because these tasks have diverse domains?
>
> **Response:** Thanks for the comment! Indeed, we follow the previous work on OOD text classification [4-6] to evaluate the sentiment analysis and natural language inference tasks.
>
> [4] Prompt-based distribution alignment for domain generalization in text classification. Jia et al., 2022.
>
> [5] Pada: Example-based prompt learning for on-the-fly adaptation to unseen domains. Ben-David et al., 2022.
>
> [6] Domain generalization for text classification with memory-based supervised contrastive learning. Tan et al., 2022.
>
> > **Weakness 2**: This paper is not very well-written and organized.
>
> We make responses to each concept of this concern as follows.
>
> > **Weakness 2-1:**  Section 3 lacks clarity, and it contains typos (e.g., "f_d" instead of "f_g" in equation (5)) and confusing notation that seems to influence the understanding of the algorithms.
>
> **Response:** Thanks for your careful check! We have carefully checked the typos and revised it as follows:
>
> $\max_{\theta_m} \min_{\theta_{f_d}} L_d (\theta_{g}, \theta_{m}, \theta_{f_d}\ ; D_{te}, D_{tr}) $  (Eq. (5))
>
> > **Weakness 2-2**: Overloading a section with excessive content can lead to confusion and make it hard for readers to follow the logic flow of ideas. It would be better if each section can address a distinct aspect of the work, e.g., a separate section of task formulation from the current Section 3.
>
> **Response:** Thanks for your suggestion! We will take it for the revision. In the current version, we organize Section 3 in a way that starts with a general overview and then delves into specifics. In particular, **Sec. 3.1** gives an overview of the **model architecture**, **Sec. 3.2** gives an **overview of the learning/optimization strategy**, and then **Sec 3.3** explains in detail the **training objectives**, and finally **Sec 3.4** gives the **full procedure of the algorithm**.
>
> > **Weakness 2-3**:  It is confusing to mention "meta-test" in Section 3.4: is it a different concept from the inner-loop in Section 3.3?
>
> **Response:** Following the previous work on DG based on meta-learning [7,8], **meta-test is an additional training process after the meta-training**. In particular, the meta-training process optimizes the bilevel optimization objective on the splits of meta-source and meta-target samples and the **meta-test process optimizes the inner-loop objective on the source samples**.
>
> [7] Learning to Generalize: Meta-Learning for Domain Generalization. Li et al., 2017.
>
> [8] Feature-Critic Networks for Heterogeneous Domain Generalization. Li et al. 2019.
>
> > **Weakness 3:** The illustration of paper can be more informative. For example, figure 2 can explain the component (d) is only used during training to help the memory network to learn the invariant representation; the current version may make the readers misunderstand it will also work in inference.
>
> **Response:** Thanks for your suggestion! we will improve Fig. 2 by adding more information about the difference across meta-training/meta-test/inference phases.
>
> > **Weakness 4:** Both meta-learning and memory networks have been used before in addressing the out-of-domain text classification. What is the most motivating thing (i.e., novelty) about this work? It would be better if the authors can explain why the approach is a significant contribution over previous methods, and what the implication is for future work.
>
> **Response:** Thanks for your constructive comment and suggestion! We highlight the **contribution/novelty** of this work compared with the previous work on meta-learning and memory networks as follows:
>
> - Previous works on **OOD classification based on meta-learning** mostly use the MAML framework [9], where the meta-learner aims to optimize the **meta-target classification loss** by learning a **parameter initialization** [10,11]. In contrast, the meta-learner in our method aims to learn a **domain invariant feature space** by learning a **memory augmentation**. **The novelty of our work is reflected in the design of meta-learning objectives (e.g., Eq. (7), Alg. 1, etc.)**. **The advantage of our design lies in the ability to leverage an additional memory network to learn more robust feature representations across domains**.
>
> - Previous research on **memory networks** primarily focuses on utilizing test information to adapt the model (refer to Section 2 in our paper). However, there have been limited studies on **memory-based out-of-distribution (OOD) text classification**. To the best of our knowledge, only one existing study [12] leverages the memories of source-domain samples to augment contrasting features for computing supervised contrastive loss. Our work differs significantly from this study [12]. Firstly, **our memory network is trainable**, whereas they employ static source-domain banks that are not optimized during training. Secondly, **we explicitly utilize memory as feature augmentation to enhance invariant representation learning**, whereas they employ memory as contrasting features for computing the contrastive loss.
>
> [9] Model-agnostic meta-learning for fast adaptation of deep networks. Finn et al., 2017.
>
> [10] Learning to generalize: Meta-learning for domain generalization. Li et al., 2018.
>
> [11] Learning transferable features in meta-learning for few-shot text classification. Xu et al., 2020.
>
> [12] Domain Generalization for Text Classification with Memory-Based Supervised Contrastive Learning. Tan et al., 2022.
>
> > **Question A:** What are the keys in the key-value memory network (L207)?
>
> **Response:** Following the previous work on key-value memory networks [13, 14], the keys $\mathcal{K}$ are used to **compute the weighting scores for memory values**, represented as $\boldsymbol{\alpha} = \operatorname{softmax}(\mathbf{q}^\top \mathcal{K})$, where $\mathbf{q}$ represents the query vector corresponding to the current hidden state. This is equivalent to Eq. (1) in our paper. In addition, the keys are randomly initialized and optimized to select the memories for invariance learning during the training phase.
>
> [13] Key-value memory networks for directly reading documents. Miller, A., et al., 2016
>
> [14] Large memory layers with product keys. Lample et al., 2019.
>
> > **Question B:** What is the memory size in Figure 5? Is it the number of keys?
>
> **Response:**  The memory size in Fig. 5 denotes the **number of memory values** $\vert \mathcal{V} \vert$ (L212) in the K-V memory network.
>
> **Yes**, it is equal to the number of keys, which is used to ensure the computation of memory in Eq. (2).
>
> > **Question C:** How can we reach the claim in L476-479 from Figure 3?
>
> **Response:** We explain in detail the Fig. 3 as follows. **Fig. 3 (a)** illustrates the feature space of **baseline model w/o memory augmentation**. **Fig. 3 (c)** illustrates the feature space **with the memory augmentation**. We can clearly observe that **Fig. 3 (c)** shows stronger feature matching between the target domain and source domains compared with **Fig. 3 (a)**. Furthermore, we can quantitatively calculate the $\mathcal{A}$-distance [2] between the target domain and source domains, and obtain that **0.527 in  Fig. 3 (a) v.s. 0.093 in  Fig. 3 (c)**. This shows that our method can effectively learn memory augmentations to achieve domain invariance in the feature space.
>
> **Thanks again for your effort in giving detailed and constructive comments. If you have more questions, feel free to discuss them with us. We are looking forward to your further messages in the discussion period.**

---

### Official Review · Reviewer_goM8 · 2023-08-13

**Soundness:** 4

**Excitement:**

3: Ambivalent: It has merits (e.g., it reports state-of-the-art results, the idea is nice), but there are key weaknesses (e.g., it describes incremental work), and it can significantly benefit from another round of revision. However, I won't object to accepting it if my co-reviewers champion it.

**Paper Topic And Main Contributions:**

The paper investigates the problem of out-of-domain classification with the aim of extending a classifier, trained on multiple source domains, to a new target domain. The main challenge comes from the disparity between source domains and the target domain. The paper proposes to address this challenge via the use of memory-augmentation to enhance the quality of the domain-invariant embedding space. Experiments on sentiment analysis and natural language inference tasks with ROBERTA_base show promising improvements of the proposed method over the baselines.



**Questions For The Authors:**

- There is a potential missing baseline: [1] Guo, H., Pasunuru, R., & Bansal, M. (2020, April). Multi-source domain adaptation for text classification via distancenet-bandits. In Proceedings of the AAAI conference on artificial intelligence (Vol. 34, No. 05, pp. 7830-7838). Have the authors considered the relevancy of this paper?

- Have the authors tried their method with more challenging datasets/scenarios when the target domain is distant from the source domains?

**Reasons To Accept:**

- The paper is well written with clear motivation and description of the methodology, experimental results and analyses.
- The underlying intuition for the design of the architecture and the learning process is well justified and easy to understand.
- The experiments are comprehensive and support the claims of the paper.

**Reasons To Reject:**

- There is a potential missing baseline: [1] Guo, H., Pasunuru, R., & Bansal, M. (2020, April). Multi-source domain adaptation for text classification via distancenet-bandits. In Proceedings of the AAAI conference on artificial intelligence (Vol. 34, No. 05, pp. 7830-7838).
- The discussion on how the two datasets mentioned in the paper are challenging for out-of-domain classification should be provided. This will better highlight the contribution the the proposed framework.

**Reproducibility:**

3: Could reproduce the results with some difficulty. The settings of parameters are underspecified or subjectively determined; the training/evaluation data are not widely available.

**Reviewer Confidence:**

2: Willing to defend my evaluation, but it is fairly likely that I missed some details, didn't understand some central points, or can't be sure about the novelty of the work.

---

> ### Author Rebuttal · Authors · 2023-08-28
>
> ### Dear Reviewer goM8,
>
> Thank you for taking the time to read our paper and giving constructive comments. We hope our point-to-point responses can address your concerns.
>
> > **Weakness 1**: There is a potential missing baseline: [1] Guo, H., Pasunuru, R., & Bansal, M. (2020, April). Multi-source domain adaptation for text classification via distancenet-bandits. In Proceedings of the AAAI conference on artificial intelligence (Vol. 34, No. 05, pp. 7830-7838
> **Question 1**: Have the authors considered the relevancy of this paper [1]?
>
> **Response:** Thanks for your valuable comments! The work by Guo et al. [1] proposes a DistanceNet model for multi-domain adaptation, which uses a mixture of five domain distance measures, including $\mathcal{L}_2$, cosine, MMD, Fisher and CORAL, to minimize the domain distance between the target domain and source domains.  They further employ a multi-armed bandit controller to dynamically select the sequence of source domains to deliver the best outcome on the target domain task.
>
> Since in the OOD setting considered by our work, the target domain is unseen during training, it is impossible to optimize a multi-armed bandit to select the sequence of source domains in the training phase. Thus, **we take the idea of DistanceNet [1] to use a mixture of domain distance measures to learn the invariant feature representations across source domains**. This revised baseline is named **DistanceNet-S**, which is different from the original DistanceNet model [1] in that **we use the distance measures across source domains** instead of the distance measures between the target and source domains. **Loss function** of  the **DistanceNet-S** baseline can be represented as:
>
> $\mathcal{L} = \sum_k \alpha_k \sum_{i,j} D_k(S_i, S_j) + \ell_{task}$, where $D_k(\cdot, \cdot)$ represents different distance measures from {$\mathcal{L}_2$, cosine, MMD, Fisher, CORAL} and the weighting coefficients $\alpha_k$ are tuned by the source-domain dev sets.
>
> We *reproduce* the baseline **DistanceNet-S** [1] and compare it with our method on six datasets. The results that **DistanceNet-S** [1] optimizes a mixture of distances to obtain invariant representations across source domains and **achieves better results on 5 of 6 datasets**. **Our method** uses a **meta-learning strategy** with a novel **memory-based invariance learning method** to learn the **invariant feature representation between the target domain and source domains**  and achieves the best results on six datasets.
>
> | Method | Amazon | IMDB | SST-2 | MNLI | SNLI | SICK |
> | ---------- | ---------- | ---------- | ---------- | ---------- | ---------- | ---------- |
> | RoBERTa (*baseline*) | 92.0 | 90.1 | 88.3 | 79.2 | 78.1 | 61.5 |
> | DistanceNet-S [1] | 92.8 | 92.0 | 90.7 | 78.5 | 80.2 | 62.8 |
> |**our method**| **94.0** | **93.5** | **92.4** | **83.0** | **82.3** | **65.7** |
>
> [1] Multi-source domain adaptation for text classification via distancenet-bandits. Guo et al, AAAI 2020.
>
> > **Weakness 2**: The discussion on how the two datasets mentioned in the paper are challenging for out-of-domain classification should be provided. This will better highlight the contribution of the proposed framework.
> **Question 2**: Have the authors tried their method with more challenging datasets/scenarios when the target domain is distant from the source domains?
>
> **Response:** Thanks for your valuable comments! We analyze the challenges of the used datasets for out-of-domain (OOD) text classification by **cross-domain evaluation**. We investigate the performance of a baseline text classification model (RoBERTa-base)  trained on the dataset when tested on out-of-domain examples. As listed in the following table, **a significant drop in performance when tested on out-of-domain data compared to in-domain data indicates the dataset's difficulty for generalization**.
>
> | Testing Scenario | Amazon | IMDB | SST-2 | MNLI | SNLI | SICK |
> | ---------- | ---------- | ---------- | ---------- | ---------- | ---------- | ---------- |
> | In-domain | 95.3 | 94.9 | 93.4 | 84.5 | 88.5 | 90.3 |
> | Out-of-domain | 92.0 | 90.1 | 88.3 | 79.2 | 78.1 | 61.5 |
> | Drop | -3.3 | -4.8 | -5.1 | -5.3 | -10.4 | -28.8 |
>
> As shown in the above table, the cross-dataset scenario (testing on IMDB, SST-2, SNLI, and SICK) is challenging for OOD evaluation. In addition, we further evaluate our method on a **rumour detection dataset PHEME** [2], which consists of five domains: Charlie-Hebdo (C), Ferguson (FR), Germanwings-crash (GW), Ottawa-shooting (OS), and Sydney-siege (S). The OOD results (measured by macro-F1) by the leave-one-domain-out evaluation are listed in the below table. **The results show that the domain distance of PHEME [2] poses a significant challenge in OOD classification (75.9 for ID v.s. 61.1 (-14.8) for OOD)**. **The results also show the effectiveness of our method for OOD classification in rumour detection**.
>
> |Method | C| FR | GW | OS | S | Avg.|
> | ---------- | ---------- | ---------- | ---------- | ---------- | ---------- | ---------- |
> | In-domain (*oracle*) | *78.2* | *60.3* | *82.8* | *80.7* | *77.6* | *75.9* |
> | RoBERTa (*baseline*) | 66.4 | 43.7 | 68.7 | 64.5 | 62.1 | 61.1 |
> | +memory | 65.7| 42.6 | 71.6 | 65.7 | 63.2 | 61.8 |
> | inv. learn. (w/o memory) | 64.6| 43.2 | 71.0 | 66.8 | 64.2 | 62.0 |
> | **our method** | **68.2** | **47.6** | **76.8** | **67.3** | **68.2** | **65.6** |
>
>
> [2] Exploiting context for rumour detection in social media. Zubiaga et al., 2017
>
> **Thanks again for your effort in giving detailed and constructive comments. If you have more questions, feel free to discuss them with us. We are looking forward to your further messages in the discussion period.**

---

### Meta-Review · Area_Chair_pPBj · 2023-09-19

**Recommendation:** 4

**Metareview:**

The paper presents a memory augmentation technique to address OOD text classification. The memory is used during meta-learning to enhance the invariance of the representations. Experiments on semantic analysis and NLI show promising gain, and SotA result. The writing as a matter of presentational issue needs to be improved (fully agree with VVu8). The discussion with goM8 justified the reason behind the task selection adequately. The combination of memory and meta-learning is certainly well-justified (specially after reading the discussions).  An interesting work!

---

### Decision · Program_Chairs · 2023-10-07

**Decision:**

Accept-Main

**Comment:**

The paper presents a memory augmentation technique to address OOD text classification. The memory is used during meta-learning to enhance the invariance of the representations. Experiments on semantic analysis and NLI show promising gain, and SotA result. The writing as a matter of presentational issue needs to be improved (fully agree with VVu8). The discussion with goM8 justified the reason behind the task selection adequately. The combination of memory and meta-learning is certainly well-justified (specially after reading the discussions).  An interesting work!